# Dual-Purpose Cattle Raised in Tropical Conditions: What Are Their Shortcomings in Sound Productive and Reproductive Function?

**DOI:** 10.3390/ani13132224

**Published:** 2023-07-06

**Authors:** Carlos Salvador Galina, Mariana Geffroy

**Affiliations:** Departamento de Reproducción, Facultad de Medicina, Veterinaria y Zootecnia, Universidad Nacional Autónoma de México, Ciudad Universitaria, Mexico City 04510, Mexico

**Keywords:** cattle reproduction, peasant farming, developing countries, reproductive technologies

## Abstract

**Simple Summary:**

Dual-purpose systems are important components in the food production line, especially with the increasing interest of consumers demanding food products that are not produced in intensive enterprises that are, in the eyes of the public, deleterious to animal welfare. How would biotechnologies such as artificial insemination or embryo transfer help to improve the efficacy of small community farmers, as the dominant system in the tropics? Different limitations must be considered, such as inadequate record keeping, the limited ability of farmers to invest in their farms, an undefined breeding program limiting genetic progress, challenging environmental conditions, inadequate infrastructure, nutritional practices, and health issues; these are the main considerations of this review.

**Abstract:**

Dual-purpose husbandry might well be the most commonly employed cattle management system in tropical regions worldwide. The advantages of producing both meat and milk, although in reduced quantities, gives an edge to the farmer in coping with the volatile economic conditions that prevail in the region. Herein, we discuss the different methods of cattle management under tropical conditions based on the financial and social structure of this system. An account of the sanitary and nutritional conditions available to the farmers and how these factors affect the profitability of the enterprise will also be given. Finally, we will discuss how these systems can take advantage of several biotechnological procedures, and how these tools (such as controlled natural mating, artificial insemination, and embryo transfer) affect reproductive outcomes. The present review will mainly concentrate on production systems located less than 1000 m above sea level, as the problems and shortcomings of cattle raised above this arbitrary landmark are quite different.

## 1. Introduction

The world population is expected to reach 9.7 billion in 2050 [1]. The need for animal products will increase by 50% in the next 30 years due to the increasing numbers of people and consequent purchasing power in developing countries [2,3]. Cattle plays a significant role in economic development and food security worldwide. Still, at the same time, their husbandry generates about 70% of the total livestock greenhouse gas emissions on the planet, polluting not only air, but also water and soil, with negative consequences for biodiversity [4,5]. Thus, there is a need to improve existing cattle farming systems to make them more sustainable and productive, while maintaining high animal welfare standards. Dual-purpose systems, defined as those wherein meat and milk are produced simultaneously, based on grazing as the main feed source, are the most frequently employed livestock enterprises in the tropics [6,7]. Production is lower in dual-purpose husbandry compared to systems wherein milk or meat are produced by specialized breeds. In some regions, such as Latin America, the dual-purpose system is well defined, with a preference for producing milk, leaving the income from beef production for times of economic scarcity [8]. In tropical regions of Africa and Asia, it seems that small dairy farm holders using dual-purpose cattle favor milk production [9]. Conversely, some farmers prefer meat production, leaving milk for local consumption. Dual-purpose farms contribute to regional meat and milk production and benefit farmers and communities by creating jobs and contributing to daily earnings and food security, thereby preventing migration from rural areas [10].

However, there is a need for continuous evolution in dual-purpose management practices, in those countries situated in the region, to meet the demand for more and better products from local cattle without damaging the environment [11]. In addition, there is fierce competition for water and other resources between the need for industrial growth and the requirement to procure feed for the ever-increasing population, particularly in developing countries [12]. So, the sum of these issues, in the face of the dire need to procure food for the human population, poses a serious question for the future. Traditionally, increases in animal production have been addressed by increasing animal density on the farm, with the unfortunate consequences of an increase in greenhouse gas emissions affecting the environment [13,14].

## 2. Data Recording and Economical Aspects

It has been observed in several tropical cattle production systems that record keeping is, at best, inefficient [15]. For example, González-Padilla et al. [16] observed that 49.1% of Mexican dual-purpose smallholders kept no records, 47.7% had handwritten records, and only 2.8% kept them on a computer. Furthermore, in Colombia, record keeping depends on the size of the production system, where 91% of large livestock producers (over 251 animals) kept records compared with 41% of very small producers (those with fewer than 30 animals) [17]. Thus, production decisions are primarily based on experience, hunches, and the needs of the farm, but the lack of records acts as a barrier to improving efficiency and profitability [18]. Veterinary services are usually geared towards clinical work and have little interventions in recording data. In previous efforts to elucidate the value of the sparse records available, Anta et al. [19] and Galina and Arthur [20] encountered an apparent bias because farmers tend to discard the data of unproductive animals. Additionally, the inaccuracies of records on paper are readily apparent. Baca Fuentes et al. [21], in a large sample of records kept on cards, found that 22% of the animals diagnosed as pregnant by AI were out of the normal range of 261–290 days, suggesting that many of those cows became pregnant by natural mating, which followed the AI program. Similarly, Fuentes et al. [22] reported that almost 30% of the records analyzed were erroneous and presented calving intervals out of the normal range. There is ample information in the literature reporting calving intervals longer than 15 months, and this figure accounts only for animals actually calving [23,24]. For this reason, studies such as that of Bishop and Pfeiffer [25] observed that in a typical Rwandan dual-purpose cattle production system in Africa, no records are kept, and management decisions depend on direct observations to approximately calculate age at puberty, postpartum anestrus, and calving efficiency. Modeling the impact of various management strategies that farmers apply on their farms, via interviews, has been employed to discern possible pitfalls in production systems [26,27]. In general, the significant obstacles continue to be very poor health and nutrition, which translate into less-than-optimal reproductive efficiency. In summary, using physical records as the main database has provided an erroneous view of production efficiency in such regions, as these data only represent the events that occurred (for example, calving), but leave aside all the animals that remained barren and later discarded from farms. Furthermore, Mwanga et al. [28], in an extensive survey in East Africa, found that the majority of farmers who use AI kept records, purchased more animal feed, and conferred larger benefits to labor by hiring workers whose average wages were higher than those using bulls for breeding. Thus, a dichotomy exists between how much the farmer is able to invest in producing more animal products and their economic limitations.

Potential solutions to the record dilemma do exist, and tailor-made computer software for record keeping in tropical conditions has become available and has been adopted by livestock farmers in recent years [15,29]. Information storage capabilities have come a long way, along with the education of the farmers, which encourages them to invest time in data entry and analysis (Figure 1). Promoting record keeping in farms and monitoring productive, reproductive, and economic parameters is essential to ensure the success of technology transfer and ameliorate assistance for farmers [17]. This will ensure that in the future, researchers will have a clearer picture of the actual performance of dual-purpose cattle.

While there are numerous and various systems of production in the lowland tropics, a common predictor of success in all of them is the farmer’s financial ability to invest in the farm. Based on these premises, Galina et al. [10] proposed a random division of producers into subsistence, medium-income, and those with sound budgets. As this paper observed, the subsistence farmer will always have a role as a provider of small quantities of milk or meat to the local community or to their families [30]. There have been many international and national programs to alleviate the situation of small community farmers, producing variable results. Unfortunately, most farmers are reluctant to implement changes that will affect their everyday routine because of ignorance, fear, labor shortages, and lack of economic, land, and water resources [31]. Moreover, if the program is subsidized, farmers are usually aware, due to previous experiences, that there will be no continuity when the project finishes. On the other hand, higher-income farmers will take more risks and adopt technology because they have better access to information, savings, and land ownership [32]. In addition, the influence of gender on technology adoption has to be considered. Villarroel-Molina et al. [26] found that although women were not community leaders, they were open to communication and rapidly adopted new technologies to improve the reproductive efficiency of the cows. These aspects of female management resulted in an increased productivity of 20% compared to male farmers.

The number of smallholders willing to invest in their farms is increasing rapidly in the tropics, particularly those with mixed systems in which manure is used to fertilize the crops and other feedstuffs to nourish their livestock [11,33]. Meat or milk produced under these more natural pasture conditions has attracted a large portion of consumers who are willing to purchase products from these types of systems; they are perceived to be more sustainable, environmentally friendly, and natural, and to feature more humane production conditions for livestock [34]. An interesting example of how consumers shape the market is shown in the study of Duncan et al. [35] in India and Ethiopia. They found that enhanced market quality was associated with improved dual-purpose systems. Using AI as an example, they discovered that in India, the management system improved in a competitive market. In contrast, in Ethiopia, the expenses involved in an AI program were not justified, because there was a low demand for livestock products in the area. In general, medium-income farmers have organized themselves in cooperatives to reduce production costs. However, sometimes, the individualistic spirit of some farmers in the tropics has impaired the development of these enterprises [36]. 

## 3. Genetic Considerations

The efficiency of a livestock scheme is derived from the responses of a particular genotype of the animal to its environment. The genetic structure of the animals will determine their capability to cope with environmental stressors, such as climate, endemic diseases, or parasitism. How the animal reacts to these stressors will depend on its ability to use the nutrients available to promote growth, production of either milk or meat, and fertility. 

In an earlier comprehensive study in Venezuela, Vaccaro et al. [37] reported that crossbred animals between *Bos indicus* and *Bos taurus* are more suitable for the environment than cattle of European breeds such as Holstein or Jersey. This study confirmed that, with rare exceptions, exotic breeds will always underperform in tropical regions in which harsh environmental conditions prevail. For this reason, Cunningham [38] indicated the advantage of producing better milk or beef through crossbreeding beef breeds with dairy cattle, thereby making an F1 animal. Crossbreeding of cattle raised in tropical conditions has been used extensively either with a purpose in mind, i.e., breeding two pure-breed types to produce an F1 animal, or indiscriminately crossing several breed types with no breeding strategy [39]. The latter practice results in an assortment of cattle locally known as indigenous breeds or mestizos. No matter which system is used, there are a few examples where crossbreeding does not positively impact production. The creation of F1 animals improves production parameters due to hybrid vigor. However, the selection of breeding F1 females either with F1 males (inter se mating) to produce an F2 generation, or backcrossing with one of the parental breeds, has not produced the expected results because the benefits gained from heterosis of 50% are not sustainable [40]. Another constraint that farmers face is the lack of sufficient animals to implement a sophisticated breeding program; thus, they are left with no choice but to use whatever bull is available, and lose the advantages that an organized breeding program can bring.

Maiorano et al. [41] compared selected Gyr cattle either for growth traits or for milk production, observing a clear genetic differentiation between dairy and beef animals, illustrating the need for more studies of this type, at least on selected populations of dual-purpose cattle. Maciel et al. [42], working in South Africa, confirmed previous data which showed that genotype and environmental interactions affected the reproductive performance of Nguni cattle. 

Keeping purebred bulls on the farm, under the conditions of small farm holders, is not possible or sustainable due to the costs of purchasing and maintaining these animals. Farmers have therefore employed hybrid bulls to reduce costs, as males can be easily substituted and sold as beef. However, several reports have indicated that crossbred animals can render poor results in reproductive performance. Moraes et al. [43] compared purebred Hereford with Bradford bulls (a synthetic crossbred), and found the former to be more efficient as sires. In an extensive survey, Chacón et al. [44] reported that crossbred animals fail to achieve an adequate score compared to pure breeds. These results demonstrate that crossbred bulls may be less productive in a breeding program. 

## 4. Environmental Issues

As dual-purpose systems are located mainly in the tropics, they are directly subjected to environmental conditions such as photoperiod, temperature, humidity, and diseases, which directly influence production in adverse ways.

Heat stress is defined as high temperature and relative humidity levels combining to the point at which they exceed the thermal comfort zone. It has been found to be one of the leading causes of reproductive problems in cattle. In this case, the animal is unable to regulate homeostasis and other physiological activities correctly [45]. Adaptation to heat requires the coordinated action of behavioral, physiological, neuroendocrine, and molecular mechanisms. The most used indicator to measure heat stress is the temperature humidity index (THI). In cattle, a THI higher than 73 is considered mild heat stress, while 90 or more is considered severe heat stress [46]. In dual-purpose systems, the genetic makeup plays a role in the ability of cattle breeds to adapt to heat and humidity, with Bos taurus being more affected by heat stress than Bos indicus breeds. The latter animals are capable of physiological and cellular adaptations to achieve lower metabolic rates, and have developed more effective heat loss mechanisms to maintain a stable body temperature [47]. 

In cows, heat stress has numerous effects on reproduction. It alters the onset of ovarian activity, compromising follicular function and oogenesis, estrus duration and intensity, and the display of overt signs of estrus [46]. Díaz et al. [48] observed that high THI levels in Bos indicus cows prolong the resumption of ovarian activity postpartum, and animals are less likely to express estrous behavior. In the case of estrus duration, Zebu cattle presented an average estrus of 12.4 h if the temperatures were below 27 °C, and in higher temperatures, it lasted for 9.3 h [49]. Additionally, Orihuela et al. [50] showed that under a continuous observation regime, the percentage of animals detected with overt signs of estrus in the hottest hours of the day can be as low as 30%, compared to an increase of 60% in the activity shown during the night or in the cool hours of the early morning. In a review, Wolfenson and Roth [51] found that the effects of heat stress on reproductive hormones are the reduction of estradiol secretion that impairs estrus duration, and LH secretion, which causes insufficient growth of preovulatory follicles, ovulation, and the formation of a suboptimal corpus luteum. Lamothe-Zavaleta et al. [52] reported that cortisol secretion is higher in the rainy season in Mexico, characterized by higher temperatures. Elevated cortisol acts as a stressor, inhibiting the response of the pituitary gland to GnRH, delaying the chain of events leading to estrus, ovulation, and, consequently, fertility. In the same study, there were changes in the number of days of the peak of estradiol and LH, together with an earlier decrease in progesterone concentrations during the warmer season. Moreover, the production of insufficient progesterone has been found in cows exposed to chronic and seasonal heat stress, thereby reducing fertility [53].

The effects of heat stress in bulls have also been documented, resulting in lower sperm quality, reduced sperm motility, and an increased percentage of abnormal sperm [54]. Nichi et al. [55] observed more sperm defects in bulls during the summer than during the winter under tropical conditions in Brazil. The effects depended on the breed of the bull, where Bos indicus bulls presented fewer sperm defects than Bos taurus. Similar results were reported by Leite et al. [56]. 

Current projections of climate change state that the world will reach the 1.5 °C threshold increase of human-induced warming by 2040 [57]. This will result in an increase in extreme weather events such as higher temperatures, changes in precipitation, and prolonged drought [58,59]. Livestock production in extensive systems is one of the most threatened food-producing industries, as the elevated temperatures will negatively impact feed and forage quality and production, water availability, and disease frequency. Dual-purpose systems are quite vulnerable to climate change, which directly affects available resources, productivity, and finances [60]. Gallardo-Chávez et al. [61] conducted an interesting survey in which small dual-purpose farmers on the west coast of Mexico were interviewed. The research approach was based on a structured questionnaire, capturing the perception of farmers to climate change. Part-time farmers with another source of income in services or commercial endeavors were less vulnerable to climatic variations and extreme weather events. This is not the case with farmers whose only sources of income are agriculture and livestock production. 

Another environmental factor linked to reproductive problems in dual-purpose systems is the presence of infectious diseases. The humid and hot conditions in the tropics provide the environment for multiple parasites, bacteria, protozoans, and disease vectors to thrive and facilitate pathogen transmission [62]. Gastrointestinal nematodes, ticks and tick-borne diseases, reproductive viruses, and bacteria are some of the most commonly found illnesses in the tropics; they affect cattle production and therefore local economies. Of all these diseases, tropical cattle present a high prevalence of both internal and external parasites. The main internal parasites that cause disease and major problems in ruminant livestock in the tropics are *Strongyles*, *Toxocara*, *Trichuris*, *Fasciola*, *Paramphistomum*, *Monezia*, *Dictyocaulus viviparus* and *Eimeria* [63,64]. However, infections are almost always caused by a mixture of species [65].

The main strategy for cattle parasite control in the tropics is the use of antihelmintics, insecticides, antiprotozoal agents, and ectoparasiticides. Often, the indiscriminate use of these chemicals has resulted in the development of resistance in parasite populations, thus impacting production and reproduction while posing an environmental and public health risk. For example, in México, at least 71% of cattle herds have nematodes resistant to ivermectin [66]. Sustainable control alternatives have been developed for parasite control, such as the use of biological pest control, grazing management strategies, the breeding of parasite-resistant animals and the use of phytochemicals [67,68].

Reproductive infectious diseases in the tropics cause infertility, abortions, mummifications, stillbirths, or the birth of weak calves [69]. Diseases such as brucellosis, leptospirosis, chlamydiosis, campylobacteria, neosporosis, bovine viral diarrhea, infectious bovine rhinotracheitis are prevalent in dual-purpose systems affecting productivity, limiting the access to local markets, and in the case of brucellosis, it poses a public health issue [70,71,72]. These diseases can be introduced into the farms after the purchase of cattle from infected herds; moreover, the lack of physical barriers, biosecurity protocols, isolation areas, and vaccination schemes, along with the lack of education and training in preventing disease, make dual-purpose farms vulnerable to illnesses [62,72]. Furthermore, the treatment of infected animals is more complicated in tropical regions, particularly in developing countries, due to difficulties in the provision of veterinary services and an unregulated pharmaceutical industry, both of which limit diagnosis and control [73].

## 5. Nutritional Overview

In dual-purpose systems, where animals predominantly graze in rangelands, their diets mainly comprise grasses with some supplementation [7]. The quality of pastures represents the greatest nutritional challenge in the tropics, and animals often suffer from protein, energy, and/or mineral deficiencies [74]. Usually, pasture forage is low in protein and soluble carbohydrates, highly fibrous, and has limited digestibility [75]. 

Poor nutrition is reflected in low reproductive rates. As expected, cows with a poor body condition have more problems related to infertility. If females present a body condition score of less than 2.5 (on a scale of 1–5) following the birth of their calves, they will show delayed ovulation, a sign of weak estrus, and will need more services to achieve conception. When cows show a body condition score of 3 to 4, these problems do not occur. Heifers with low body condition scores are likely to have delayed puberty, reduced udder health, lower conception rates, and chronic malnutrition, leading to anestrus [76].

Romanzini et al. [77] showed that positive results occurred consistently when appropriate nutritional forage was fed. Overall, economic return and profitability were better than when herds were fed low-quality forage. Hence, financial returns vary between years and among supplements. In a survey carried out in Malawi, Banda et al. [78] concluded that when management was erratic, particularly concerning feeding practices, housing, and health issues, reproductive performance in the herds of small farm holders was diminished. Borges et al. [79] reported on the effect of convincing farmers to implement strategies of dietary supplementation in their herds in a trial carried out for 18 months in Venezuela. Supplementing with by-products available in the area had a positive impact on production.

Dietary supplementation has been reported as a possible solution to increase the intensity and duration of the overt signs of estrus [80]. Bottini-Luzardo et al. [81] supplemented crossbred animals with *Leucaena*, a tropical legume, but found inconclusive results on the value of such an approach. In that study, the intake of Leucaena affected the number of cows exhibiting ovarian cyclicity and extended corpus luteum life, but did not affect follicular development and estrous behavior. Earlier, Soto et al. [82], comparing supplemented and non-supplemented cattle presenting a marginal body condition after calving, reported that concentrate supplementation favored body weight gain but not the onset of ovarian activity. In other studies, monitoring body weight during late gestation seems to be a valuable tool to predict early reproductive success postpartum. Díaz et al. [83] reported that the resumption of ovarian function was not affected by the combination of calf separation and progestogen treatment when cows were in an acceptable body condition in the last trimester of pregnancy. In further reports, Díaz et al. [83] concluded that dams in good metabolic condition during the latter part of gestation achieve higher pregnancy rates regardless of the time postpartum when estrous cyclicity begins. In a related study, Jolly et al. [84] reported that a low body condition score at calving was associated with prolonged postpartum anestrus. However, when the dams maintained their live weight, ovarian cyclicity resumed within 50 days. These studies call attention to the fact that the environmental conditions at each site affect the success of a reproductive program. 

The adoption of alternative and more sustainable husbandry systems has increased in the tropics, where small farm holders have begun to employ silvopastoral systems because of their multiple benefits [85]. In this system, trees, shrubs, and pasture grow in the same area, including both the use of legumes and/or perennial trees that can be utilized as alternatives to pastures. This management strategy increases the dry matter available per square meter, while protecting soil from degradation [86]. Silvopastoral systems are perceived to be better for animal welfare as they provide shade [87]. Animals in these environments are less prone to disease, and there is an improvement in the nutritional quality of the feed [88]. These improvements influence reproduction through nutrition. When a portion of the concentrate was replaced with foraging inside a silvopastoral system in Costa Rica, there was no difference in follicular growth and fertility with cows that only grazed in the pasture [89]. Producers may economize by using this system and therefore reducing external purchases of concentrate and feed, while maintaining productive and reproductive parameters. More studies are needed in this area.

In summary, dietary supplementation in dual-purpose cattle undoubtedly has a beneficial effect both on production and reproductive rates. The overall economic feasibility of this model in tropical regions remains to be tested.

## 6. Breeding Procedures

### 6.1. Natural Mating

Dual-purpose systems generally employ sires/bulls as their primary source to mate females. There is very little research on the fertility obtained with this system, although based on data gathered on natural mating programs in the tropics [24,90], one would not expect pregnancy rates to rise over 80%, even if all the female animals in the farm are cycling, and thus at risk of becoming pregnant. Reports on reproductive efficiency in dual-purpose systems show an important variation within the scientific papers consulted (for an earlier review, see Galina and Arthur [91]). Galina et al. [24], in a review of bull performance in breeding programs, concluded that basing our judgment on the reproductive capacity of bulls, which is built into pregnancy rates in a herd, can be misleading, as there could be several females remaining acyclic during the breeding program. This latter issue may dictate the bull to female ratio and whether a single or multiple sire system could be put in place. Smallholders might not have the possibility of multiple sire programs, as their infrastructure may not permit the implementation of such a program. However, it is important to remark that the lack of sound reproductive records, together with the bias of including only cows impregnated following a natural mating program in the calculations, provides limited information about bull performance.

Chacón et al. [44], in an extensive survey carried out in Costa Rica, found that almost 30% of the males tested were unsuitable for breeding in herds mostly dedicated to extensive beef cattle enterprises. There is, surprisingly, no documented evidence of how dual-purpose farmers select their sires for natural breeding programs. Most likely, the test used during the selection is the actual reproductive performance of the bulls instead of a breeding soundness evaluation [92]. If a male manages to make females pregnant on the farm, then the animal is selected as a sire for future breeding programs. Another common approach is to purchase bulls from stud farms with the understanding that the males will be sound for reproduction. In addition, Lassala et al. [93] reported that bull soundness evaluations were carried out by only 31% of farm holders at the time of sire purchase. Afterwards, bulls were rarely evaluated, before the breeding season or annually. 

### 6.2. Artificial Insemination

#### 6.2.1. Semen Production and Handling

In the case of more progressive farmers willing to use AI, there has been a tendency for farmers in the tropics to utilize the professional services of a veterinarian to collect the semen of their sires onsite, and subsequently transport the samples to their laboratories. In a previous report, Hernandez et al. [94] found that the longer it took to process the semen after collection, which was extended in a nutritious diluent for preservation during transportation, the more the quality of the product deteriorated. There is a paucity of data concerning appropriate sources of semen preserved in tropical settings, as few companies are dedicated to commercializing semen from native breeds. 

Argudo et al. [95], in a study carried out in Ecuador aiming to preserve the Creole breed in the high altitudes of this country, compared two types of diluents and two methods of semen collection (electroejaculation and artificial vagina), confirming previous data which showed that the former collection method is not as suitable for semen preservation [96]. The alternative that most of the progressive farmers in the area tend to utilize is to purchase semen produced commercially.

#### 6.2.2. Estrous Detection

The detection of overt signs of estrus can be a daunting task in the tropics, because climatic conditions influence the animal behavior during estrus, and observed behaviors are sparse and short [97]. Additionally, in many dual-purpose farms, the animals are kept tied up for most of the day in shabby shelters with wet, messy floors (Figure 2), and in the case of cattle raised in extensive systems, they might disperse and have limited time for interaction (Figure 3). Moreover, animals are sent to pasture after milking, and spend the rest of the day outdoors, unsupervised. Galina and Orihuela [97] in an extensive review on the subject, concluded that the existing social interactions in the herd could jeopardize the common signs of estrus (allowing a cow to be ridden by other peers). Animals may display estrous behavior even when a pharmacological device is in place, or when ovarian structures such as large follicles are present. Farmers characterized as dual-purpose would rely in some instances on the behavior of each cow, as knowledge on their animals and their tendencies is crucial for detecting animals in estrus. Because of these shortcomings, the pharmacological control of estrus is emerging as an alternative [98]. However, despite an array of protocols to synchronize estrus and ovulation, and the use of fixed-time artificial insemination, THI, and the season, either wet or dry, can directly affect the reproductive performance of dual-purpose cattle [99,100]. Additionally, there is a marked difference when applying a pharmacological approach to heifers [101] compared with postpartum cows, as the latter will be influenced by the presence of the calf, either via suckling, or via the milk let down during lactation [102,103]. In effect, complex interactions between seasons of the year, the age of the dam, supplementation strategies, and an array of calf separation techniques and/or hormonal treatments could jeopardize the interpretation of published information, which only accounts for one or two variables. Economic adjustments between dietary supplementation, milk yield, and reproductive performance have yet to be determined in milking cows, either with the calf present or absent.

#### 6.2.3. The Role of the Technician

The application of semen into the genital tract of a female via a catheter inserted into the body of the uterus has long been a very common practice in intensive dairy systems; technicians utilize the technique almost daily. This gives them an advantage compared with an individual whose interventions with the technique may be sporadic. In the case of DP cattle, the number of cows inseminated each day can be minimal, and the duties of a technician can also be extended to the milking of animals, in addition to other chores on the farm. The dexterity developed by the individual in charge (using which the technician has to introduce a catheter through the cervix) can be problematic, as there are reports that Bos indicus genotypes often present a tortuous or c-shaped cervix, making it rather difficult to introduce the catheter in the body of the uterus [104,105].

To counter the lack of continuity in applying AI, farmers have introduced methods such as placing a flag by the entrance of the farm, so the technician dedicated to AI only covers the given area [106], or in a farming community, naming one person to serve the other areas through a type of communal arrangement. No matter what method is used, the technicians require periodic retraining. Peters et al. [107], compared 20 professional inseminators with herders also in charge of AI and, using a syringe which was located via ultrasonography, found that neither type was accurate in the placement of semen in the body of the uterus, with high variability occurring in each group. In contrast, Müller-Sepúlveda et al. [108], working with small community farmers, reported differences in the age and experience of the inseminator, wherein individuals with more than 10 years of experience achieved a higher pregnancy rate than inseminators with 1–4 years.

The introduction of fixed-time AI has created various problems for the inseminator, as the pressure to place the catheter in the appropriate site is a major constraint. Russi et al. [109] interviewed inseminators with particular reference to their activities outside their work. Among other things, they found that confidence in the insemination technique was necessary, as the inseminator who did not believe in the efficiency of fixed-time AI obtained lower pregnancy rates than those who trusted the technique. Additionally, inseminators that underwent retraining achieved better results. Oliveira et al. [110] found that the sequence of insemination after simultaneous thawing of semen straws affected conception rates following timed AI. However, there was a startling effect on fertility, depending on the sire utilized, which needed further research. Proper hygienic measures are critical for the success of the AI technique, as shown by Bas et al. [111], who reported increased fertility in cows inseminated using a protective sheath covering the AI gun, compared to peers without a protective sheath.

### 6.3. Embryo Transfer

In a recent review, Contreras et al. [112] pointed out several shortcomings associated with applying embryo transfer in small community farmers. The advantage of placing F1 embryos in F1 recipients is that it would eliminate the need for crossbreeding cattle, which is difficult and time-consuming (see the section on Genetics). However, other factors have an important effect on the success of the technique. One such factor is the increasing evidence that Bos indicus embryos are structurally different from Bos taurus embryos [113,114]. Thus, the grading of an embryo as a suitable candidate for transfer can be erroneous [115]. In earlier studies, Aguilar et al. [116] found as much as a 30% error in the grading of Bos indicus cattle using a stereoscopy microscope. This disparity was not evident in Bos taurus embryos [117]. A recent communication about the morphological and physiological dissimilarities between Bos taurus and Bos indicus embryos [118] indicates that the latter cells are more prone to damage via freezing. More work is needed in a detailed study on the morphological characteristics of F1 (Bos taurus x Bos indicus) embryos if these are to be used on an industrial scale to facilitate heifer replacement programs in the tropics.

The success of embryo transfer (ET) programs on dual-purpose farms in tropical conditions depends on several factors, such as the characteristics of the donor and recipient cows and the bulls used for offspring production [119]. The embryos can be obtained directly from the donor (in vivo) or by fertilizing the oocytes collected through ovum pickup (in vitro). The oocyte yield, maturation, and fertilization vary depending on various factors such as bovine subspecies, age, breed, and environmental conditions [120,121,122]. A study by Slade Oliveira et al. [123] suggested that using Gyr cows instead of Holsteins to recover oocytes for embryo production may increase pregnancy rates in tropical environments. To ensure a successful program, the number of recipient cows prepared to receive an embryo and confirmed pregnancies after transfer are crucial indicators [124]. Recipient cows must go past the event of maternal recognition to ensure embryo survival and the maintenance of the pregnancy. Crossbreeding between Bos taurus and Bos indicus cows and embryos can reduce embryonic mortality related to limited nutrient availability in tropical and subtropical climates compared with Bos taurus alone [125]. Additionally, hormonal treatments have been beneficial in facilitating self-appointed ET programs in the tropics [126], with synchronization protocols used to prepare recipients, being theoretically able to reduce embryonic death caused by an underdeveloped endometrium [127,128,129]. A study by de Oliveira Bezerra et al. [130] showed that fixed-time AI and ET have similar pregnancy rates in crossbred cattle, suggesting that the latter may be feasible under tropical conditions. For instance, this technology has been proposed to alleviate reduced fertility during heat stress [131,132,133]. Nonetheless, earlier studies have not given such optimistic results. For example, Montiel et al. [134] had to select recipients out of 215 cows, of which only 97 were found to be suitable for receiving an embryo. The reported overall pregnancy rate was around 25%. Similar results were reported by Alarcon et al. [124]. Additionally, Sánchez et al. [135] carried out an economic feasibility study of developing embryo transfer programs with small farm holders, concluding that due to the small numbers of animals used either as donors or recipients, the cost exceeds the benefits of using ET in this sector of tropical cattle production. 

### 6.4. Influence of Milking and Calf Management on Reproduction

In dual-purpose systems, the milking process takes place every day. Usually, this process happens just once, in the early mornings [136]. In several characterization studies in different parts of Latin America, it has been observed that most farmers use manual milking [137], wherein the cow is accompanied by her calf (Figure 4). The calf is usually allowed to suckle on residual milk, one teat, or other combinations [138]. This facilitates the process of milk let down, but has a strong influence on the resumption of ovarian activity, where the effects of suckling, the presence of the calf, and nutritional status all participate [139]. Cow–calf interactions directly affect anestrus, where the more encounters between the mother and the calf and the higher suckling stimulus, the more profound/prolonged the anestrus will be [140]. When suckling happens more than twice daily, plasma LH concentrations decrease, and ovarian inactivity rises [139]. 

The optimal calving interval in dual-purpose farms is 365 days, meaning that in the best conditions, the cow has to become pregnant 85 days after calving [97]. Several cow–calf separation methods currently take place in the tropics to help cows reactivate ovarian activity; these are partial separation (or fence-line), early weaning, temporary weaning, restricted suckling, and the use of nose-flaps [103]. A detailed description of these methods can be found in the review by Orihuela and Galina [103]. Knowledge of how the behavior and physiology of the animals affect these aspects is very important, without neglecting the aspects of wellbeing, both in the mother and in the offspring [103].

In addition, one of the advantages of calf suckling is the lower incidence, prevalence, and relative risk of both clinical and sub-clinical mastitis, which ameliorate udder health, and as a consequence, improve reproductive efficiency [136,141,142]. Besides being beneficial for the cow, there are benefits for their offspring as well; the presence of the dam is speculated to offer a protective effect for both respiratory diseases and scours in calves [143], and to improve general health through increased immunoglobulin absorption from the colostrum [144]. Healthy cows and calves with increased welfare management turn into productive animals, which bring profits to small community farmers.

In contrast, fewer farmers use automatic milking systems (Figure 5), which need less time, effort, and use of human labour. Calves in these systems will be fed artificially, usually without contact with the dam. Thus, they will not influence the resumption of ovarian activity. 

## 7. Conclusions

Several factors influence reproduction in the tropics. There is a need to integrate these factors to attain high productive and reproductive standards, while maintaining high standards of animal welfare.Research that bears in mind the economic, cultural, and educational level and the infrastructure on farms is needed in tropical environments, particularly regarding dual-purpose cattle, to ameliorate reproductive and productive conditions.The adoption of technology in the tropics is related to the ability of the extension services to adequately transfer the technology, which emanates from research, to the farm holders.

## Figures and Tables

**Figure 1 animals-13-02224-f001:**
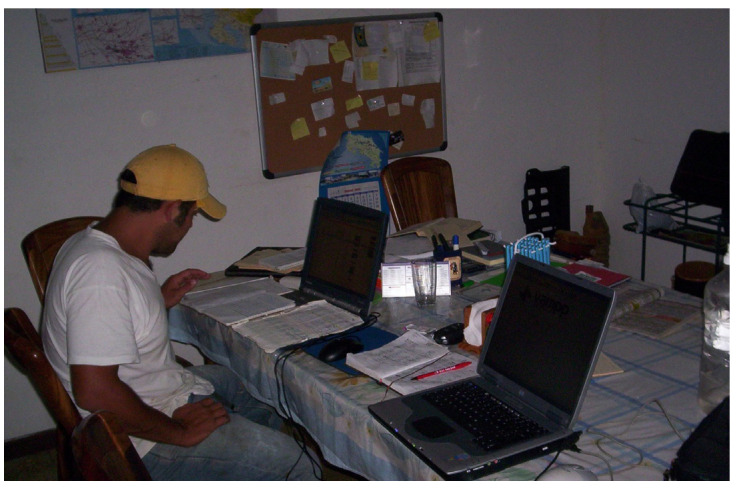
Farmer transferring information from paper to a computerized program for record keeping.

**Figure 2 animals-13-02224-f002:**
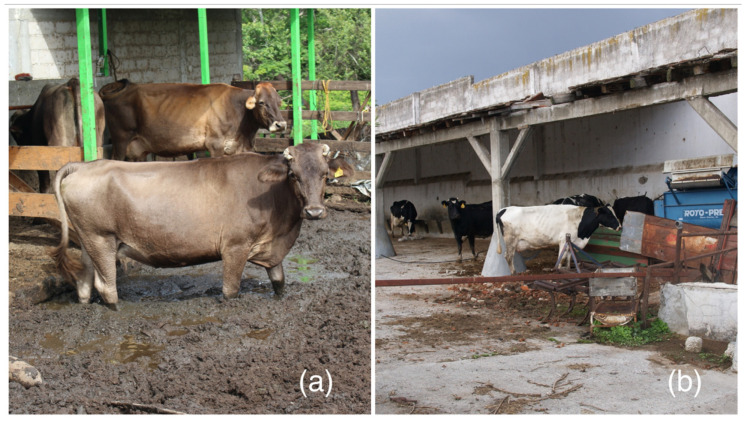
Dual-purpose cows in unsuitable facilities. (**a**) The appalling conditions of the flooring, the poor maintenance, (**b**) and the presence of rubbish will negatively impact the presentation of normal behaviors.

**Figure 3 animals-13-02224-f003:**
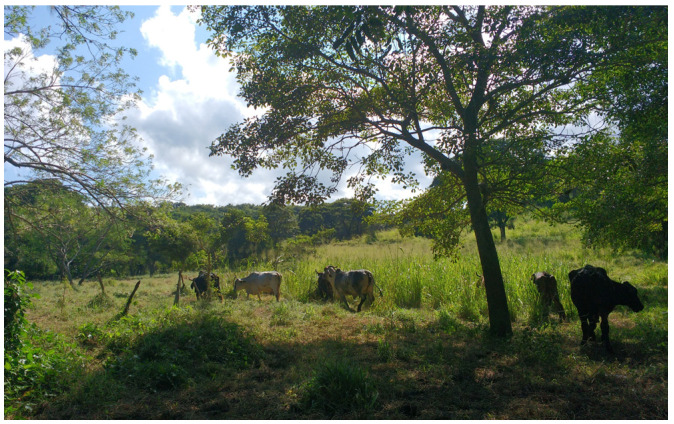
After milking or management, cows in extensive systems are left by themselves to graze and feed on pasture, hindering the task of estrus detection.

**Figure 4 animals-13-02224-f004:**
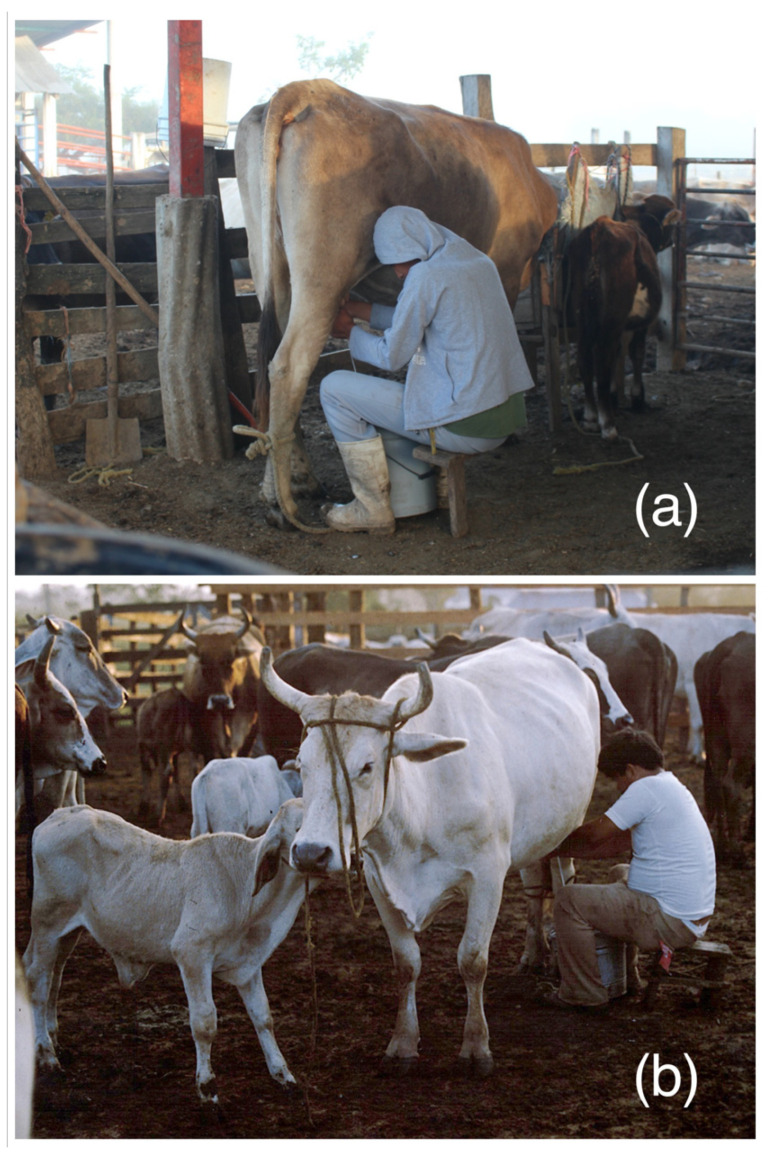
(**a**) Hand-milking is still a popular method in dual-purpose systems, in which the presence of the calf is necessary to facilitate milk let down (**b**).

**Figure 5 animals-13-02224-f005:**
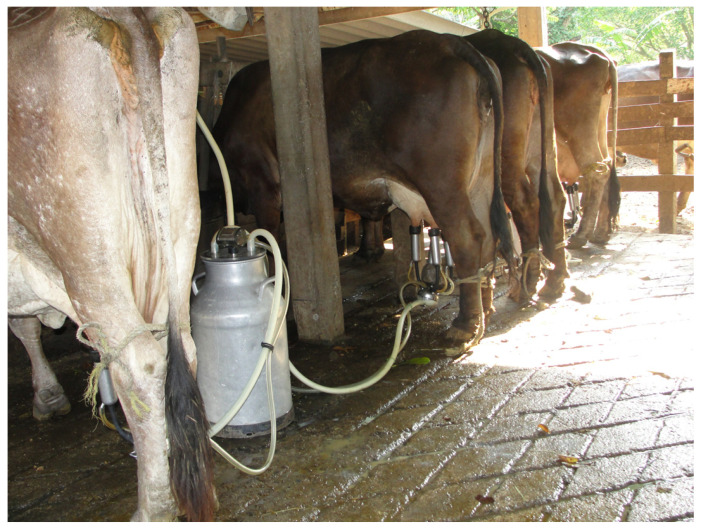
Use of automatic milking systems in clean and suitable facilities.

## Data Availability

No new data was created or analyzed in this study. Data sharing is not applicable to this article.

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
