# Peer review of "Dual-Purpose Cattle Raised in Tropical Conditions: What Are Their Shortcomings in Sound Productive and Reproductive Function?"

_animals, 2023, doi:10.3390/ani13132224_

Round 1

Reviewer 1 Report

The manuscript is fairly written, however, there are some major concerns. The title of the manuscript is "Reproductive function of dual purpose cattle raised under tropical conditions" and the authors have a literature review on different random topics, and which one of them includes reproduction.

The authors describe the need for improvement in dual purposes management, but there is a huge confounding of demographic and dual purposes livestock. Another major concern is that there is no material and methods on the manuscript. In a literature review, the authors need to describe how the papers for the review were selected and what words were used for the search,  criteria to select the manuscripts, etc. ? It isn't clear on "general considerations"  the reasons that the authors choose random topics other than reproduction.

The English is fairly good, however there are some sentences and words that will need an attention.

Author Response

We have changed the title as we agree with the reviewer as it does not reflect the essence of the paper. “Dual purpose cattle raised under tropical conditions. What are their shortcomings for a sound productive and reproductive function?”

As we pointed out in the review, searching for adequate examples of studies published in accessible databases is not an easy task, as researchers documenting information on dual purpose cattle, to our knowledge, are few.  As the document was already too long, we did not feel it was appropriate to include a material and methods section, as indeed, we had to use several searching strategies to identify the documents that could be useful for the structure of the review. We have also used data of either Bos taurus or Bos indicus, to illustrate the points we wanted to raise. We also indicated in the manuscript, that we limit our comments to cattle raised in the lowland tropics. We thought that for the interest to the reader, presenting an overall picture of the many shortcomings, hindering the full productive potential of the animals, was necessary to underline the existing factors prevailing in the lowland tropics.

Reviewer 2 Report

I have appreciated this large review dedicated to the difficulties linked on dual purpose cattle in under tropical conditions. The main factors have been correctly presented.  Again we can observed that education is still needed to see an improvement.

I have only two smalls remarks on line 39 and 63

39 Do you have some general data collected in your under tropical conditions who can be used to better define a dual purpose cattle.

63 data recording. I believe that the the regular system of visits by a vet or a zootechnician is also a factor who can influence the quality of the data

Author Response

Thank you for your comments here below you can see the reply for both. 

Reply for comment about line 39; Indeed, a very pertinent comment. By putting together the review, we noticed how difficult it is to define “dual purpose cattle” as there are entities whose primary goal is the production of milk and the offspring is raised and sold as it is carried out in “specialized dairy systems in the temperate zones”. This decision is based on the price of milk as for example in Costa Rica, the liter of milk is almost double than in Mexico. The same goes for producers whose main objective is to raise steers for the market, as the price of milk is not economically attractive, thus the product is used in different schemes (sold to the local community, to the middleman, to companies such as Nestlé or as a supplement to young steers). 

Reply for comment about line 63; We have added this sentence to the text “Veterinary services are usually geared towards clinical work and have little interventions in recording data”.

Reviewer 3 Report

The review needs to be reorganized and AI and ET sections need a better description in their utilization, benefits and problems in dual-cattle in the tropics more than in the general knowledge of the technology, particular problems in particular farming systems. A detailed review is enclosed in a separated document.

Reproductive function (performance?) of dual-purpose cattle raised under tropical conditions.

Authors: Carlos Salvador Galina, Mariana Geffroy1

Simple Summary:

Line 9: Replace are becoming, by are important components

Line 11: Replace on the welfare of the animal, by to animal welfare

Comment: The review has a wider scope than the role of AI and ET in dual-purpose systems; accordingly, please rewrite the paragraph to balance the contribution of each topic considered in the review. In addition, give another view to the title. I am not sure whether performance is the right word, but I think that function suggests a more physiological approach.

Abstract:

Line 25: Please delete the factors influencing cattle reproduction in tropical

Introduction

Line 31: Replace in a growing world, by The world population is expected…

Lines 34-39: Please rewrite the paragraph.

Line 42: Replace are the exclusive products of the enterprise by are produced by specialized breeds

Lines 53-57: Please rewrite the paragraph.

Comment: Introduction needs a better focus on the main topic. The impact of dual-systems animal husbandry on greenhouse gas emissions is reduced as >80% of cattle inventory is located in India (>40%), Brazil, China, and Europe. Eventually, it is better to consider gas emission balance (emission = capture) and to approach environmental topics avoiding underscoring the importance of animal production in the world food security. Finally, no aims are considered in this section to understand the organization of this review.

2. General considerations Please replace the title.

Comment: This section requires a better title. Authors are describing the dual-purpose systems in tropical environments, and probably, the main factor affecting their performance. What about the breeds involved in dairy and beef production?

2.1 Data recording

Lines 64-65: Please review this sentence as below, authors suggest that recording depends on the size of the herd (Colombia, 91% of units above 250 cows keep records), etc. I suggest separating small farmers from larger and to describe their reality.

Lines 75-78: There is little difference between 77% and 81% to criticize the accuracy of recording system. In addition to a wrong assessment, mating after AI has little impact on decision taking.

Lines 89-93: It is required a more general view, considering countries, herd size, etc.

Lines 95-103: Please reorganize the paragraph considering first Lines 99-103, and then, 95-97. Lines  97-99 can be deleted.

Comment: I suggest approaching this topic considering first the cultural background behind, and then herd size and government support to small farming, that most of the time  is directed to self-maintenance that commercial purposes (a social than an economic topic).

2.2 Economies (I suggest replacing it by economical aspects)

Lines 110-111: Please replace purchasing power, by just financial ability to invest in the farm

Line 129: It is not clear what modest and more significant budgets means.

Comment: I suggest a better link of the economic status of farmers with the reproductive performance of dual-purpose herds. Alternatively, I suggest fusing 2.1-2.3 in a general description of dual-purpose farming context with the current subtitle of the section or something similar.

2.3 Genetic progress (without records?)

Line 149: Delete optimal

Line 152: Please precise the crossbreed (taurus-indicus? Dairy and beef?)

Lines 166-169: This comment is too academic, as the approach of farmers to recording data does not consider this analysis.

Line 176: Replace enterprise by farm or herd

Line 178: Replace heterosis by hybrid as smallholders don’t care about the genetic background of bulls.

Lines 184-189: It needs to be consistent with the section on records.

Comments: I suggest fusing the first three sections considering socio-economical description of farmers (distribution between subsistence, medium-income and large), stocks and breed (including regions, tropical America is similar to Africa or Asia?). Authors should give a general view of cattle farming in tropical environment.

2.4 Environmental considerations

Lines 202-203: Please replace:  in the impact of heat stress, where the consequences in dual purpose systems will always affect Bos taurus breeds more than Bos indicus by in the ability of cattle breeds to adapt to heat and humidity, being Bos taurus more affected than Bos indicus breeds to heat stress. Heat stress is a clinical condition.

Line 204: Please replace the development of by have developed

Line 217: Replace finding by found

Lines 217-220: Please adjust the paragraph to highlight the effect of heat stress on the pulsatile secretion of LH that affects terminal follicular development, estradiol secretion and estrous behavior.

Line 232: Please move bulls after Bos indicus.

Lines 234-247: Interesting paragraph but it looks out of scope. As the review is intended to describe the current status of dual-purpose cattle farming, I recommend deleting it.

Comment: Compared to heat stress, the impact of infectious disease was less developed, in particular parasites diseases that affect seriously Bos taurus compared to Bos indicus breeds in the tropics. I recommend a bit more attention to this point.

2.5 Nutrition (considerations?)

Line 272: Please replace Due to undernutrition, by As expected,

Lines 273-277: Please develop a more mechanistic view of the influence of nutrition on reproduction (just as heat stress). Energy balance has a strong influence on lactation and reproduction after partum, including embryo losses, and so protein balance at mating.

Comment: It is required a wider description of feeding for dual-purpose cattle production in the tropics, just to understand how it is affecting reproductive management and performance there. Is there any seasonal curve of production of natural forages? Does rain season affect reproductive strategies and performances? When authors refer to silvo-pastoral practices, what sort of trees are they referring to, etc.

2.6 The bull (Please consider a separated section, like Reproductive Strategies or something like that, and then refer to natural mating and then AI)

Lines 330-332: Both sentences appear contradictory. Please rewrite this paragraph.

Line 334: Please use a clear marker of fertility.

Lines 335-337: Please develop this point

Lines 339-340: Sentence is not clear.

Line 346: Replace simple observation by actual reproductive performance of the bulls, but observations are usually linked to this assessment.

Line 347: Replace to fertilize by to pregnant

Line 348: Replace Another alternative by Another common approach (it is a quite common approach to get males)

Lines 349-350: Delete this sentence, it is out of scope

Comment: Authors need to describe better the bull use for mating in the tropics (male-female ratio, bull individual or group mating and their performances, smallholders and larger herds, bull reproductive assessment, etc.).

2.6.2 Semen production and handling (this topic falls in the section of AI)

Comment: I recommend deleting this section as it doesn’t contribute with interesting or particular information.

2.7. The application of artificial insemination (I recommend replacing this point by a section of reproductive strategies (natural mating,  artificial insemination, and ET)

2.7.1 Estrous detection

Line 374: Replace enterprise by farms

Line 375: It is a very critical view, for probably a very exceptional case (tied stalls are not common in smallholder in temperate regions).

Line 380-381: Delete a technique known and or insemination by appointment as it is well-known concept, fixed-time AI or FTAI, no one use insemination by appointment.

Lines 384-385: Delete either by suckling or by the milk let down and add lactation to close the sentence.

Lines 394-396: Are there picture of adequate housing for dual-purpose cattle in the tropics?

Line 399: Replace impeding the task of by hindering or someone else

Comment: Estrous detection description requires a wider and more descriptive view; it is a too critical view for a management that may apply just to small farming.

2.7.2 The role of technician

I recommend its deletion as a separated section.

Comment: The section of AI also requires a better description. The topic needs to be reviewed, and both sections (estrous detection and technicians) should be considered as part in the main topic (has not relevance as separated points). There is no information on the use of AI, the role that fulfills, the organization in large farms and small farming in the tropics, and the main benefits and problems that AI has for dual-purpose cattle.

2.8 The use of embryo transfer

Comment: Again, the general knowledge of procedures associated to embryo transfer should not be part of this review. As in the AI section, authors should focus on the use of ET in cattle under tropical climate and under different farming systems.

2.9 Influence of milking and calf management on reproduction

Lines 483-485: This sentence needs to be supported by references

Comment: This section needs to be placed after the Nutrition section.

Conclusions

Line 525: Replace DP by dual-purpose

Lines 527-529: Please in addition to focus the description of AI and ET utilization to this conclusion, consider the degree of use in large and small farms.

References:

Please Rearrange the references to follow the Journal format. 

Author Response

Thank you for your comments, please see the attachment.

Round 2

Reviewer 1 Report

The authors have made a significant amount of changes on their manuscript, which had a significant improvement. However, a structured review should use the same structure as research articles and ensure that they conform to the PRISMA guidelines. Methods should be added in the manuscript as recommended by the guidelines (i.e., eligibility criteria, information sources, search strategy, selection process, etc.)

Author Response

Dear Reviewer: Thank you very much for your prompt reply. We considered that reviewers two and three were satisfied with our corrections consequence of their constructive criticisms.

We have constructed the review based on our first title, “REPRODUCTIVE FUNCTION OF DUAL PURPOSE CATTLE RAISED UNDER TROPICAL CONDITIONS.” For our original search, we used three major databases, “WEB OF SCIENCE, SCIENCE DIRECT, AND SCOPUS” using the keywords “DUAL PURPOSE CATTLE.” Over 2000 references were obtained, so we decided to be more specific using “ DUAL-PURPOSE CATTLE RAISED IN TROPICAL CONDITIONS”. From this search, three major journals stood up “TROPICAL ANIMAL HEALTH AND PRODUCTION” with 653 records, “LIVESTOCK PRODUCTION SCIENCE” with 80 results, and “LIVESTOCK SCIENCE” with only 8 records. Also, there were a considerable number of abstracts and full papers in Spanish or Portuguese, so we undertook another search in the Latin American databases “REDALYC. ORG, SCIELO, AND PERIODICA”. Also, in some instances, we used the bibliography cited in the articles that were finally selected for the review, to expand the information we needed to construct the article. Finally, we used our own published information to complete areas in the manuscript.

Halfway from analysing our data, we used the same databases explained earlier, with a more specific search string based on the section of the paper, for example, “NUTRITION AND DUAL PURPOSE CATTLE”.

We are aware of the existence of the PRISMA methodology, in our opinion, we think it is too late in the process to make a report using the guidelines (to know exactly the number of articles found, the number of references that were discarded by their abstract and body and the number of articles that made it to the final cut). We also think that even though we did not use the PRISMA methodology per se, the information presented in the article has scientific value and comes from reliable sources.

In addition, any review will have a bias based on the opinions of the authors, where some factors might receive more attention than others. It is also acceptable that criticism emanating from what was written is possible. We intended to write a story based on our experience and others that touched the subject and were useful for us to include in the manuscript.